# Tele-Rehabilitation for People with Dementia during the COVID-19 Pandemic: A Case-Study from England

**DOI:** 10.3390/ijerph18041717

**Published:** 2021-02-10

**Authors:** Claudio Di Lorito, Carol Duff, Carol Rogers, Jane Tuxworth, Jocelyn Bell, Rachael Fothergill, Lindsey Wilkinson, Alessandro Bosco, Louise Howe, Rebecca O’Brien, Maureen Godfrey, Marianne Dunlop, Veronika van der Wardt, Vicky Booth, Pip Logan, Alison Cowley, Rowan H. Harwood

**Affiliations:** 1Division of Rehabilitation, Ageing and Wellbeing, School of Medicine, University of Nottingham, Nottingham NG7 2UH, UK; mszleh1@exmail.nottingham.ac.uk (L.H.); ntzro1@exmail.nottingham.ac.uk (R.O.); maureengodfrey47@gmail.com (M.G.); mariannedunlop@icloud.com (M.D.); Vicky.Booth@nottingham.ac.uk (V.B.); pip.logan@nottingham.ac.uk (P.L.); 2Lincolnshire partnership NHS foundation Trust, Lincoln LN1 1FS, UK; carol.duff3@nhs.net (C.D.); carol.rogers3@nhs.net (C.R.); jane.tuxworth@lpft.nhs.uk (J.T.); Jocelyn.bell1@nhs.net (J.B.); rachael.fothergill@nhs.net (R.F.); lindsey.wilkinson2@nhs.net (L.W.); 3Division of Psychiatry and Applied Psychology, School of Medicine, University of Nottingham, Nottingham NG7 2TU, UK; alessandro.bosco@nottingham.ac.uk; 4Abteilung für Allgemeinmedizin, Präventive und Rehabilitative Medizin, Philipps-Universität Marburg, 35037 Marburg, Germany; v.vanderwardt@uni-marburg.de; 5Institute of Care Excellence, Nottingham University Hospitals NHS Trust, Nottingham NG5 1PB, UK; alison.cowley@nuh.nhs.uk; 6School of Health Sciences, University of Nottingham, Nottingham NG7 7TU, UK; Rowan.Harwood@nottingham.ac.uk

**Keywords:** tele-rehabilitation, dementia, physical activity, exercise, COVID-19

## Abstract

Introduction: The Promoting Activity, Independence and Stability in Early Dementia (PrAISED) is delivering an exercise programme for people with dementia. The Lincolnshire partnership National Health Service (NHS) foundation Trust successfully delivered PrAISED through a video-calling platform during the Coronavirus Disease 2019 (COVID-19) pandemic. Methods: This qualitative case-study aimed to identify participants that video delivery worked for, to highlight its benefits and its challenges. Interviews were conducted between May and August 2020 with five participants with dementia and their caregivers (*n* = 10), as well as five therapists from the Lincolnshire partnership NHS foundation Trust. The interviews were analysed through thematic analysis. Results: Video delivery worked best when participants had a supporting caregiver and when therapists showed enthusiasm and had an established rapport with the client. Benefits included time efficiency of sessions, enhancing participants’ motivation, caregivers’ dementia awareness, and therapists’ creativity. Limitations included users’ poor IT skills and resources. Discussion: The COVID-19 pandemic required innovative ways of delivering rehabilitation. This study supports that people with dementia can use tele-rehabilitation, but success is reliant on having a caregiver and an enthusiastic and known therapist.

## 1. Introduction

Dementia presents with a cluster of symptoms, including impairment in motor skills [1,2,3,4]. Keeping physically active is beneficial for executive functioning, mobility, activities of daily living, independence, and quality of life of people living with dementia [5,6,7,8,9,10,11,12,13,14,15,16,17,18]. Several physical activity and exercise intervention programmes have been developed for people with dementia, targeting different outcomes. The Finnish Alzheimer’s Disease Exercise Trial (FINALEX) [13] evaluated the effectiveness of intense and long-term physical exercise on the physical functioning and mobility of home-dwelling participants living with Alzheimer’s disease. The Dementia and Physical Activity (DAPA) trial [14] evaluated the effectiveness of moderate to high intensity exercise training on the cognitive performance of participants living with dementia.

The Promoting Activity, Independence and Stability in Early Dementia (PrAISED) study intervention consists of an individually tailored programme of physical, dual-task exercises, and functional activities of daily living delivered in participants’ homes by a multidisciplinary team including physiotherapists (PTs), occupational therapists (OTs), and rehabilitation support workers (RSWs). The clinical and cost effectiveness of PrAISED is being evaluated in a randomised controlled trial (RCT) involving 368 participants [19].

In March 2020, the United Kingdom (UK) government implemented measures to slow the spread of Coronavirus 2019 (COVID-19) [20,21,22], including recommendation for people over 70 years of age and with pre-existing conditions (e.g., dementia) to self-isolate [23,24]. As a result, the PrAISED multidisciplinary teams were unable to visit participants in their homes. Therefore, the participants who were receiving the intervention were supported by therapists remotely (phone or video calling), in line with current guidance for practice [25,26]. The PrAISED research team provided the therapists with plans and guidance on how to deliver PrAISED remotely (Appendix A, Appendix B and Appendix C). The Lincolnshire partnership National Health Service (NHS) foundation Trust (LFPT) PrAISED therapy team delivered the intervention through a video-calling platform named Q Health.

Tele-rehabilitation has been successfully used in other medical areas [27], and evidence is mounting on the potential benefits in a population living with dementia [28,29,30,31]. However, there is paucity of research around the experience of delivering and receiving tele-rehabilitation for people with dementia during the COVID-19 pandemic. Given the potential need in the future to deliver services remotely (e.g., in the context of social distancing requirements), this study aims to present preliminary evidence on tele-rehabilitation for people with dementia and identify the type of clients with dementia tele-rehabilitation works for, how, and under which conditions; the benefits for the clients receiving tele-rehabilitation and the therapists delivering tele-rehabilitation; and the challenges of tele-rehabilitation and how to potentially address these.

## 2. Materials and Methods

This study abides by the consolidated criteria for reporting qualitative research (COREQ) [32].

### 2.1. Setting

Q Health (https://qhealth.io/, accessed 10 January 2021) is an NHS Digital and NHS England-approved video patient consultation solution developed by a centrally funded supplier, introduced in PrAISED by the LFPT team at the end of March 2020, as part of the COVID-19 response. The platform allowed therapists and participants to set up and attend digital appointments during lockdown. Q Health required that the user had access to technology (i.e., internet connection and a computer, tablet or smart phone) and was able to download the Q Health application, to book the digital appointments and connect for the session. The therapy team developed a simple set of instructions and supported PrAISED participants via the phone to download and set up the Q Health application. Once the application was installed, the participants were provided with a one-time use code to grant secure access, select an appointment time, and enter the video consultation.

### 2.2. Participant Inclusion/Exclusion Criteria

Participants with dementia were eligible to participate if they met the following inclusion criteria: aged 65 years or over; had a diagnosis of mild cognitive impairment or dementia (of any subtype, except Dementia with Lewy Bodies); had a Montreal Cognitive Assessment (MoCA) [33] score of 13–25 (out of 30); had an informal caregiver who was willing and able to be a participant in the study too; were able to walk without help, communicate in English, and see and hear; had sufficient dexterity to perform neuropsychological tests, mental capacity to give consent to participate, and consent to do so; were involved in the intervention arm of the PrAISED RCT at the time of recruitment (May 2020) and were being supported through Q Health.

Participants with dementia were ineligible to take part if they had a diagnosis of Dementia with Lewy Bodies or a co-morbidity preventing participation (e.g., severe breathlessness, pain, psychosis, Parkinson’s disease or other severe neurological disease); were part of the control group in the PrAISED RCT; were part of the intervention group in the PrAISED RCT but were not supported through Q Health (e.g., phone only).

Participants with dementia and caregivers were purposively sampled to represent variance in gender, relationship to each other (e.g., spouses, parent–child) and residence status (i.e., living together or independently). All the LFPT therapists delivering PrAISED through Q Health were also recruited in the study.

### 2.3. Data Collection

The participants with dementia and their caregivers were invited by the main researcher (CDL) to take part in two semi-structured qualitative interviews. After identifying potential participants through the PrAISED RCT database, CDL made a preliminary phone call to provide information about the study and register their potential interest in taking part. If the participants agreed, CDL would give them the option to have the interview either by phone or via video calling, independently or separately of the caregiver. A phone interview session was then scheduled. Before the interviews, CDL would send the participants a study information sheet and a copy of the consent form, explaining that consent would be taken orally on the day of the interview. The first interview was conducted one/two months after implementation of Q Health and the follow-up interview four months after implementation.

The therapists delivering PrAISED through Q Health were identified through the study database. CDL approached them by email and invited them to take part in a qualitative semi-structured interview, either over the phone or via video calling. Upon a positive response, a session was scheduled. Before the interviews, CDL would send the therapists a study information sheet and a copy of the consent form, explaining that consent would be taken orally on the day of the phone interview. Therapists’ interviews took place three/four months after implementation of Q Health.

All interviews were conducted by CDL, and based on a topic guide (Appendix D and Appendix E) developed in collaboration with two Patient and Public Involvement (PPI) contributors with experience of caring for someone with dementia, who were also involved in the development of the study and its protocol, and in the writing of this manuscript. While the topic guide focused on the overall experience of the PrAISED RCT, a flexible approach was used in the interviews to explore and capture information relating to Q Health. All the interviews were carried out through speakerphone, so that participants and caregivers could both hear and respond to the questions. The use of speakerphone also enabled CDL to record verbal consent and the interview session through an encrypted password-protected digital audio-recorder, as per ethical approval received by the Bradford Leeds Research Ethics Committee (Reference 18YH/0059). Data collection continued until “conceptual density” (i.e., a sufficient depth of understanding of the domains under investigation) was reached [34].

### 2.4. Data Analysis

The transcripts were downloaded from the audio-recorder, anonymized, and passed to a professional agency for transcription. The transcripts were then imported in NVivo [35] and analysed through deductive thematic analysis [36] (i.e., using study objectives as the themes). Two authors (CDL and AB) extracted the relevant selections from all the interview transcripts independently of each other and categorised them into the themes. If any discrepancies between the two authors emerged, a decision was made by involving a third author (RHH). The PPI contributors and therapists provided feedback on the findings.

## 3. Results

Five participants with dementia and their caregivers (*n* = 10) were involved in this study (Table 1). They all opted for phone interviews and agreed to have the follow-up interview. The average length of the interviews was 28 min (range 17–56). Five therapists were involved in this study (Table 1). Two were RSWs, two OTs and one a PT. They were all female and opted for a phone interview. Their interviews lasted on average 38 min (range 22–58). Quotes are reported below by identifying the participants’ and therapists’ ID (as per Table 1).

### 3.1. Type of Participant This Platform Works for, How and under Which Conditions

The qualitative interviews found that Q Health worked better with some participants than others. Therefore, using tele-rehabilitation would depend on assessment of a person’s physical, as well as digital ability:

“I don’t think all of your service could be delivered remotely, but actually tailoring it depending on what you’ve assessed with your patient, including their digital ability. Ideally, clients would have a fairly good balance and strength and they’d be able to follow instructions”.T16, PT

It was agreed that there might be risks in progressing exercises or activities through video consultation for clients who might be at risk of falling:

“With patients where their mobility isn’t as good, I would worry about being able to progress their exercises over Q Health”.T16, PT

Given the risk of injury with vulnerable clients, a condition under which tele-rehabilitation worked better was the presence/support of someone in the home during the video calling. The presence of a supporter in the home was also key for positive risk-taking:

“When I start walking backwards and counting backwards, I go a bit unsteady. So I need somebody there. Without anybody here it would not have been as beneficial”.P3042, male

Most participants agreed that the caregiver was key for successful video coaching, as they would facilitate the set-up and help to operate the system. In cases when the participant lived independently, arrangements had to be made for a supporter to be present and initiate the video call. Some pre-requisites were also needed on the therapists’ side. Creativity was deemed to be a key element to balance out some features missing in tele-rehabilitation. To promote engagement, the therapist had to show commitment and enthusiasm for the new technology:

“With the remote side of things, you have to really believe and put across to the participants that this is going to work”.T12, OT

To be able to work more effectively through tele-rehabilitation, another requirement was having previously established good rapport with the client. This was perceived to be instrumental both for participants to trust the therapist, and also for the therapist to confidently work toward progression targets:

“I had already met her before, so we just adapted and got on with it. It was more straight forward and reassuring”.C3036, female

### 3.2. Benefits for the Participants and the Therapists

A number of benefits occurred for both the clients and therapists as a result of video consultation compared to telephone support only. One benefit was seeing each other, which enhanced rapport and connectedness and made it possible to grasp non-verbal cues:

“I think it (Q Health) is brilliant. We can talk and I can see if she is laughing at something I have said and laugh along”.P3031, female

Another benefit of the video calling was that the participants could go through their exercises with the therapist, by modelling, in real time, the moves and positions. Similarly, they could engage in visually based cognitively challenging tasks:

“We do all the exercises together. She does them with me. I put my iPad where she can see me and she has got her computer where I can see her”.P3031, female

Compared to telephone support, the visual feedback also aided therapists in making a more accurate assessment of participants’ improvement (or lack thereof), facilitating progression (as opposed to mere maintenance) of participants’ goals. It was also instrumental in boosting clients’ motivation and clinician’s confidence to progress:

“It gives me confidence to progress their exercises because I can see if my participants could stand on one leg for 20 s and I can see that they’ve got a work surface next to them if they need to hold on”.T16, PT

“It was a while before Q Health was set up and I can honestly say I didn’t like it when we were doing it over the phone. I think that just seeing her makes a difference”.P3031, female

Some therapists reported that another advantage of video calling was that it was more efficient than face-to-face contact. Some therapists reported that there are rural areas where clients are hard to reach by community support. In this case, tele-rehabilitation could reduce their risk of exclusion from services. Avoiding travel to and from participants’ homes allowed for more frequent and focused sessions. The therapists also brought forward a financial argument in favour of tele-rehabilitation. They suggested that offering video support was a “good value for money” strategy to prevent participants from getting deconditioned and frail over time during “lockdown”:

“The clients that I’m supporting through Q Health would not normally be seen through an NHS service, as they would be seen as being safe at home. So these clients are not getting frail and de-conditioned because of COVID like a lot of older people that haven’t got this kind of service”.T16, PT

They continued that video calling would also prove beneficial for long-term engagement in physical activity programmes once the PrAISED programme is completed for the participant:

“A lot of clients could actually be seen remotely in the long term which may help with improving people’s compliance with increasing their physical activity levels”.T16, PT

Some benefits were also recorded for caregivers. By facilitating the video sessions, the caregivers became informed about the therapy programme and more involved in the care of the person. This could be seen as a positive and a negative as well, as the physical absence of therapists could add to caregiver burden:

“I have learned so much more about dementia in the last months. But I have absolutely no support at the moment to care for (participant), nothing at all. And my children, one lives up north and the other lives down south”.C3039, male

A benefit for the therapists was that the video calling challenged them to step out of their comfort zone, to become adaptable to the inevitable changes that COVID-19 entailed and think creatively about solutions for future practice:

“It’s just made me think of different ways I could work to make physical activity and exercise more accessible for more clients. For example, last week we wanted to make one of our participant’s exercises harder, so me and a rehab support worker, we went to the local park and we took pictures of me and her doing the exercises and then we emailed them to the participant”.T16, PT

### 3.3. Challenges and How to Potentially Address These

There were a number of challenges pertaining to tele-rehabilitation. The majority of participants felt that video calling was more valuable than no support at all or phone consultations. However, it was inherently different from home visits, where human connection occurred at a more meaningful level:

“I do miss the face to face. I just think it’s having somebody here with me, I can’t really explain it, it just doesn’t feel the same”.P3031, female

One participant explained that the digital divide between older and younger generations makes older participants with dementia less able to learn and interact through digital media:

“To see R (the therapist) on the video link feels a bit unnatural to me though, because I don’t use it (video calling) much in real life. The children are more geared to learning like that and taking things in like that than I am”.P3036, female

Therapists felt that given the limitations in environmental risk assessment typical of remote support, including relying on information reported by participants, they could not challenge participants who lived independently to the same extent they would normally during home visits:

“When somebody lives on their own you’re really reliant on them giving you a picture. And because people have their memory problems, they’re not always able to give you an accurate picture”.T14, OT

Given the remote nature of video calling, the therapists also lamented the impossibility to progress participants towards goals which required their physical presence:

“We are losing out on a lot. Like we’ve got participants who we would be going to the gym with, because they’re at the level where they can go to the gym, they want to go to the gym but we can’t”.T11, OT

Given these limitations, the therapists contended that tele-rehabilitation could be an integrated part of a hybrid delivery package, after the initial visits are (ideally) made through home visits:

“The ideal is to have at least some face-to-face contact with participants. The Royal College of OT has just published some up-to-date formal guidelines. And it makes it really clear that you must be able to assess how somebody is functioning within their home environment. But after the initial visits, if you can get a video call system that the participants can get to work then that’s really good”.T12, OT

Both caregivers and therapists agreed that Q Health did not cater to participants with dementia because of their cognitive issues. Sometimes, when the team were trying to explain to the participants how to install the programme or how to operate it, they would get very frustrated if they could not do it:

“One participant I had was tech savvy but still, she just couldn’t get hers to work and she got quite frustrated with it”.T13, RSW

Looking at potentially implementing this intervention in the community on a larger scale, the participants proposed ideas on how to address the digital exclusion that participants lacking basic IT knowledge would face, including guidance and support from therapists:

“I think most people with dementia would do panic when something different or unexpected happens. But if you have J (the therapist) on the phone and you said “Oh there is a pop up” she would just be able to say “I have seen that before, just click ok and that will be fine”.C3036, female

In recognising the importance of addressing the present digital divide, some therapists proposed ideas for making the platform more dementia-friendly:

“There are companies that produce phones specifically for older people or people who struggle with technology. And I’m just wondering if potentially they might have a very basic tablet in their range. You can set it up in a specific place in the house, have it ring at the certain time and then all they have to do is press the answer button and then they’ve got a video call and then they can hang up”.T11, OT

## 4. Discussion

This qualitative case-study contributed evidence around tele-rehabilitation to support people with dementia to remain physically active during the COVID-19 pandemic. We found that delivering the PrAISED intervention using Q Health was feasible and acceptable from the perspectives of clients, caregivers, and therapists. Similar findings have been reported by Burton and Nissen [37,38], who found that tele-rehabilitation was helpful with clients with cognitive impairments but required frequent modifications. One of the major barriers found in this study was the lack of digital literacy and access amongst clients with dementia and their caregivers. This study identified some strategies that could address digital exclusion, including ongoing support from therapists and the need to develop dementia-friendly equipment, education, and support services for users and therapists. While some attempts in developing this kind of support have been reported [39], there still is a clear need for service design, guidance, and delivery of dementia-friendly tele-rehabilitation.

Another important finding was the perceived effectiveness of delivering the PrAISED intervention using tele-rehabilitation during the COVID-19 pandemic. Although face-to-face was the preferred method, given the circumstances, the participants felt that video calling using the Q Health platform was preferable to a phone consultation.

Previous studies with cognitively impaired adults have also shown that there might be added benefits in using video—as opposed to telephone—support [37,38,40,41]. Video calling might enhance users’ satisfaction [42], facilitate the development of therapeutic alliance, which is instrumental for intervention uptake and adherence [43], and promote the empowerment of a client with dementia, who might have difficulty communicating with the therapist without face-to-face contact. From a therapist’s perspective, video calling might facilitate capturing non-verbal cues from clients.

Evidence is also mounting to the effectiveness of tele-rehabilitation, compared to face-to-face rehabilitation. The potential of cost and time efficiency of tele-rehabilitation has been noted [40]. Travelling long distances (where services cover large catchment areas) or for a long time (in the case of metropolitan conurbations) for face-to-face rehabilitation sessions is resource-intensive. Tele-rehabilitation could optimise limited time and financial resources. In terms of clinical outcomes, a non-inferiority study by Laver et al. [40] compared face-to-face versus tele-rehabilitation delivery of a programme designed to address environmental and functional issues in patients with dementia and their caregivers [44]. This study found no statistically significance difference between groups in caregiver mastery, and both groups reported significant improvements in perceptions of caring.

There are some key issues warranting careful consideration in future implementation. The creativity and enthusiasm of the therapists and service described and the recognition from the LFPT of the need to prevent participants’ deterioration serve as an illustration of what is required from a service design and set-up perspective. They also illustrate the importance of health professionals leading the way in innovations. Another crucial consideration is how to balance the practicalities of resource optimisation with the individual needs of clients. We found that face-to-face visits were felt to be better suited in the context of the initial assessment of clients’ situations (e.g., environment, falls risk, digital abilities), in establishing and in terminating support. While video consultations represent an acceptable adaptation when social distancing is required, a hybrid approach to rehabilitation would better respond to patients’ needs for effective and ethical health care.

This work is characterised by several strengths. It took advantage of an existing trial to investigate innovation rigorously in real time. This opportunistic use of data sits well with the Medical Research Council (MRC) framework [45] for the development of evidence into this new field. There are many negatives associated with the COVID-19 pandemic, particularly for older adults. In the context of the COVID-19 pandemic, the risk of social isolation and deconditioning in people living with dementia has been highlighted by Alzheimer’s Disease Support International as a significant concern, which requires ongoing support from health and social care practitioners [46]. The need for further research into technology-based support interventions for older people with cognitive impairment and their caregivers has also been identified as a research priority during the pandemic [47]. This study made an important contribution to the evidence base of tele-rehabilitation interventions to clients living with dementia. It also presented an ethical contribution to research, by giving voice to people with dementia and their caregivers, in an effort to support their ongoing involvement in research [48]. The study also featured triangulation of perspectives (through therapists’ views), which better reflects the context of the video consultations as two-way or three-way interactions.

The case study design potentially has limited generalisability. Additionally, Q Health was designed and used in one specific setting, thus having limited transferability to other video conferencing platforms. However, the intention of this study was to report a phenomenon in a specific context upon which to build further understanding. The participants included only those that used Q Health, thus excluding the views of those unable to use the system. Finally, the participants were part of an existing study (the PrAISED RCT) and therefore they did not represent the wider population of people with dementia.

## 5. Conclusions

The COVID-19 pandemic has generated the need for innovative ways of delivering rehabilitation. There is little literature about supporting people with dementia and their caregivers through video consultations. This study supports that people with dementia can use video calling, but its success is reliant on having a caregiver and an invested, enthusiastic, and known therapist. In the light of potential future situation requiring remote support or to make it an effective component of hybrid delivery of rehabilitation services, further work to make tele-rehabilitation accessible and sustainable with the most vulnerable individuals with dementia is crucial.

## Figures and Tables

**Table 1 ijerph-18-01717-t001:** Characteristics of participants.

Participant with Dementia ID	Age	Sex	Living Status	Caregiver ID	Age	Relationship to Participant	Therapist ID	Role in PrAISED	Sex
P3031	67	F	Living alone	C3031	43	Daughter	T11	Rehabilitation Support Worker	F
P3039	72	F	Living with caregiver	C3039	63	Husband	T12	Occupational Therapist	F
P3042	83	M	Living with caregiver	C3042	80	Wife	T13	Rehabilitation Support Worker	F
P3044	76	M	Living with caregiver	C3044	72	Wife	T14	Occupational Therapist	F
P3036	76	F	Living with caregiver	C3036	50	Daughter	T16	Physiotherapist	F

## Data Availability

Transcripts of the qualitative interviews are available in full from the main author (C.D.L.), upon request.

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
