# Peer review of "Tele-Rehabilitation for People with Dementia during the COVID-19 Pandemic: A Case-Study from England"

_ijerph, 2021, doi:10.3390/ijerph18041717_

Round 1

Reviewer 1 Report

  1. It is not clear from the title where (country, region) the study was conducted.
  2. The abstract needs clarification about the aims of the study, the date of sampling took place and the age of participants.
  3. Line 34-37. Please provide additional clarification of these programs.
  4. Line 52: Ref # 33- the systematic review contains little synthesis of the information. Please provide more information about the results of this review. Also how this study extends reader understanding of the topic. Authors should give specific reasons why this study is important. What does this study add?
  5. Information should not be presented in bullet point format or as a numbered list.
  6. Line 74-77: Please define the inclusion/exclusion criteria clearly?
  7. Describe semi-structured interview deeply. It is unclear whether the authors used a question guide. Data collection method should be guided by a list of topics or questions. This should be clearly included in a table.
  8. Who made interviews by phone? Are they CDL and AB only or different researchers analyzing the data? How interviews by phone were collected?
  9. How are quotes organized? The authors should consider include a table with themes, sub-themes and quotes.
  10. Is it necessary to include date/mode of interview and caregiver/dementia/therapist ID in both tables? I believe both tables are unnecessary. One table that describes the sample characteristics is required.
  11. I would suggest adding more recent references to discuss your results.
  12. Line 319: The conclusion section should be under a separate heading.

Author Response

Please see document attached

Reviewer 2 Report

This study is thought to be a valuable case study that suggested the possibility of utilizing the rehabilitation program for patients with dementia using an online system in the era of Corona-19. The following is required to be supplemented.

In the introduction, the preceding studies are simply introduced, but please describe in more detail, including the academic background related to the program.

Please present the Video-calling platform in more detail in the research method. In addition, please provide specific details about Q-health.

In the discussion, please add an academic review on the relevance of the content and effect of the study. In addition, please include the specific contents of the system.

Author Response

Please see document attached

Round 2

Reviewer 1 Report

No further comments

Author Response

Thank you

Reviewer 2 Report

Thank you.

I think it's been well modified overall.

However, please check once more in paper format composition.

And please fit the writing style of the reference.

Author Response

Thank you. We have now revised and all references  match the Journal style requirements.